# Serum IL-18/IL-13 Ratio Predicts Super Response to Secukinumab in Patients with Psoriasis

**DOI:** 10.3390/ijms26136432

**Published:** 2025-07-03

**Authors:** Dominika Ziolkowska-Banasik, Maciej Pastuszczak, Kamila Zawadzinska-Halat, Ewa Hadas, Andrzej Bozek

**Affiliations:** Department of Internal Diseases, Dermatology and Allergology, Medical University of Silesia, 40-055 Katowice, Poland; dziolkowska@op.pl (D.Z.-B.); ewahadas@interia.pl (E.H.); andrzej.bozek@sum.edu.pl (A.B.)

**Keywords:** psoriasis, super responders, IL-18, IL-13, biomarkers, precision medicine, biologic therapy, secukinumab

## Abstract

Identifying immunologic predictors of clinical responses remains an unmet need in the era of biologic therapy for psoriasis. Super responders (SRs), defined as patients achieving complete skin clearance within weeks of treatment initiation, represent an emerging clinical endotype; however, their immunological profiles remain insufficiently characterized. We conducted a prospective observational study to characterize serum cytokine profiles associated with SR status in biologic-naïve patients with moderate-to-severe plaque psoriasis treated with secukinumab, an IL-17A inhibitor. Twenty-eight patients were enrolled and stratified at week 12 into SR (PASI = 0; *n* = 9) and non-super responder (NSR; PASI > 0; *n* = 19) groups. Serum concentrations of 19 cytokines were analyzed at baseline and after 12 weeks of treatment. SRs displayed a distinct immunological signature characterized by significantly higher IL-13 and lower IL-18 baseline levels compared to NSRs (*p* = 0.002 and *p* = 0.007, respectively), alongside reduced baseline monocyte counts. L1-regularized logistic regression confirmed IL-13 and IL-18 as strong independent predictors of SR status (AUC = 0.91). Moreover, the IL-18/IL-13 ratio emerged as a highly discriminative biomarker (*p* = 0.00001, AUC = 0.86). Notably, SRs exhibited a more pronounced decline in IL-18 and IL-23 during treatment. Our findings provide novel insights into the immunopathogenesis of super response and suggest that an immunological milieu favoring Th2 polarization may promote superior outcomes with IL-17A blockade. Incorporating IL-13, IL-18, and their ratio into clinical algorithms may facilitate precision-guided biologic therapy in psoriasis.

## 1. Introduction

Psoriasis is a chronic, immune-mediated inflammatory dermatosis affecting approximately 2–3% of the global population. In recent years, a growing body of evidence has established the central role of the interleukin (IL)-23/Th17 axis in its pathogenesis. IL-23 was first characterized in 2000 as a heterodimer composed of a shared IL-12/23p40 subunit and a unique p19 subunit that is exclusive to IL-23 [1]. Functionally, IL-23 acts as a pivotal cytokine that bridges the innate and adaptive arms of the immune system. It exerts pleiotropic effects by acting on T lymphocytes and innate immune cells, including natural killer cells, macrophages, dendritic cells, and innate lymphoid cells. It plays a crucial role in the development, maintenance, and expansion of Th17 cells—key effectors in psoriatic inflammation—by promoting the production of IL-17A, IL-17F, and tumor necrosis factor (TNF). These cytokines are instrumental in sustaining epidermal hyperplasia and dermal inflammation [2,3].

Therapeutic strategies targeting this immunological cascade, particularly monoclonal antibodies directed against IL-17 and IL-23, have dramatically improved the management of moderate-to-severe plaque psoriasis. Clinical trials and real-world studies have demonstrated that a substantial proportion of patients achieve complete (PASI 100) or near-complete (PASI 90) skin clearance within the first months of treatment [4,5]. However, treatment efficacy is not universal. It is estimated that up to 15% of patients receiving IL-17 inhibitors may require a therapeutic switch within one year due to inadequate response or secondary loss of efficacy [6].

Despite the availability of multiple biologic classes, treatment decisions remain largely empirical. To date, no validated biomarkers or immunological predictors are available to guide the selection of the most effective biologic for a given patient. In clinical practice, drug choice is typically influenced by logistical, regulatory, or economic considerations rather than mechanistic precision. Consequently, therapeutic decisions often follow a trial-and-error approach, which increases healthcare costs and can delay optimal disease control, contributing to patient dissatisfaction [5,6].

In this context, increasing attention has been devoted to a distinct subgroup of patients termed super responders (SRs). These individuals achieve complete disease remission (PASI = 0) within a short period of initiating biologic therapy—typically within 12 to 16 weeks [7]. Several reports suggest that SRs may represent a distinct clinical endotype with unique phenotypic and potentially immunologic characteristics. Importantly, achieving SR status is not only associated with superior short-term outcomes, but also with prolonged drug survival and the potential for reduced dosing frequency or even treatment discontinuation without relapse, which may translate into substantial cost savings [7,8,9].

Previous studies have identified certain clinical predictors of SR status, including younger age, shorter disease duration, lower body mass index (BMI), and bio-naïve treatment history [10,11,12,13]. However, the immunological profile associated with this highly favorable response remains largely unexplored.

The aim of the present study is to perform an in-depth immunological characterization of patients with moderate-to-severe psoriasis treated with secukinumab, an IL-17A inhibitor, stratified by super responder (SR) and non-super responder (NSR) status. By analyzing serum cytokine profiles before and after treatment, we seek to identify immunologic signatures and potential predictive biomarkers associated with rapid and complete clinical response.

## 2. Results

### 2.1. Patient Stratification and Baseline Characteristics

Among the 28 patients enrolled in the study, nine achieved complete clinical clearance (PASI = 0) twelve weeks after initiating secukinumab treatment and were classified as the super responder group (SR). The remaining patients, who did not reach complete clearance (PASI > 0), were classified as non-super responders (NSR). Patients in the SR group had significantly lower baseline peripheral blood monocyte counts compared to the NSR group (*p* = 0.03). No significant differences in other clinical or demographic parameters (e.g., age, BMI, disease duration) were observed between the SR and NSR groups, as shown in Table 1.

### 2.2. Baseline and Post-Treatment Cytokine Profiles

Prior to treatment, the SR group exhibited significantly higher serum IL-13 levels and significantly lower IL-18 levels compared to the NSR group (*p* = 0.007 and *p* = 0.002, respectively; Table 2). After twelve weeks of treatment, no statistically significant differences in cytokine concentrations were observed between the groups. Treatment with secukinumab led to a general reduction in the serum levels of most analyzed cytokines in both the SR and NSR groups. Notably, IL-2 and IL-5 levels decreased following treatment in SR patients, while they increased in NSR patients. In contrast, IL-10 and IL-15 levels increased post-treatment in both groups. A significant reduction in IL-23 levels was observed exclusively in the SR group.

### 2.3. Comparison of Cytokine Dynamics Between Groups

To determine whether the cytokine changes differed significantly between groups, we calculated delta values (post-treatment minus baseline) and compared them between SRs and NSRs. Among all cytokines, IL-18 showed a statistically significant difference in delta values (*p* = 0.023), with a more pronounced decrease in the SR group. Although IL-2, IL-5, IL-10, IL-15, and IL-23 did not show statistically significant differences in delta values between groups, their changes followed the same group-specific trends as described above—with reductions primarily observed in the SR group and stable or increasing levels in the NSR group. In particular, IL-23 levels declined only in the SR group (*p* = 0.28 for between-group delta comparison; Figure 1).

### 2.4. Predictive Value of Baseline Cytokines

To assess the predictive value of baseline cytokine levels for super response to secukinumab, we performed logistic regression and stratification analysis for IL-13 and IL-18. L1-regularized logistic regression (Lasso) identified both cytokines as relevant predictors, with IL-13 showing a strong positive coefficient (+20.23) and IL-18 a strong negative coefficient (−31.13). The model showed excellent discriminative power, with an area under the ROC curve (AUC) of 0.91. Dichotomization of IL-13 and IL-18 at their respective medians further supported their predictive value. Patients with IL-13 > 0.75 pg/mL had an OR of 28.0 (95% CI: 2.07–379.27) for achieving PASI = 0. Conversely, none of the patients with IL-18 > 0.4 pg/mL achieved complete clearance, resulting in an OR of 0.0 and an undefined confidence interval.

### 2.5. Cytokine-Based Thresholds for Predicting Super Response

We used the fitted Lasso logistic regression model to estimate the probabilities of response across observed cytokine values. According to the model, IL-13 ≥ 0.39 pg/mL and IL-18 ≤ 3.90 pg/mL were the optimal thresholds beyond which the likelihood of achieving PASI = 0 exceeded 50%. This combined immunological profile may aid in stratifying patients more likely to benefit from secukinumab.

### 2.6. Correlation Between PASI and Cytokine Concentrations

Baseline PASI scores were positively and significantly correlated with the pre-treatment levels of several pro-inflammatory cytokines, including IL-12, IL-9, IL-2, IFN-α, and IL-17 (Figure 2A), suggesting broad immune activation. In contrast, no significant correlations were observed between baseline PASI and the levels of Th2-related cytokines such as IL-4, IL-5, or IL-13.

Post-treatment PASI scores were weakly but significantly correlated with post-treatment IL-5 levels (Figure 2B). Additionally, baseline PASI showed significant positive correlations with post-treatment IL-2, IL-9, and IL-23 levels (Figure 3A), while post-treatment PASI correlated positively with baseline IL-18 and IL-13 levels (Figure 3B).

In addition to PASI, we evaluated correlations between baseline and post-treatment cytokine concentrations and other clinical variables, including age and BMI. Overall, correlations were weak and did not reach statistical significance (Spearman’s ρ < 0.3, *p* > 0.05), suggesting that baseline cytokine expression was not strongly influenced by these demographic or historical factors in this cohort.

### 2.7. Correlation Between IL-18, IL-17, and Monocyte Count

Further exploratory analysis revealed statistically significant positive correlations between serum IL-18 levels and peripheral blood monocyte counts. Baseline IL-18 levels correlated positively with baseline monocyte counts (r = 0.36, *p* = 0.03), and this association was even stronger for post-treatment IL-18 levels and baseline monocyte counts (r = 0.60, *p* = 0.002). Additionally, changes in IL-18 levels (delta IL-18) exhibited a moderate positive correlation with baseline monocyte counts (r = 0.41, *p* = 0.01). These findings suggest that IL-18 expression may be partially linked to systemic monocyte activity in patients with psoriasis.

Moreover, correlation analyses revealed a weak and non-significant positive relationship between IL-17 and IL-23 concentrations, both at baseline (r = 0.27, *p* = 0.29) and post-treatment (r = 0.09, *p* = 0.74). This indicates that, within this cohort, IL-17 and IL-23 do not demonstrate a tightly coordinated expression pattern, despite their well-established functional interplay within the psoriatic inflammatory axis.

### 2.8. Cytokine Ratios as Predictors of Treatment Response

We also investigated whether cytokine ratios at baseline could predict treatment outcome (Table 3). The IL-18/IL-13 ratio was significantly lower in SR patients than in NSR patients (median 0.49 vs. 3.69, *p* = 0.00001). Logistic regression modeling confirmed its high discriminative power (AUC = 0.86), with the optimal cut-off at 0.56. Patients with ratios below this value were significantly more likely to achieve PASI = 0. The ratio also showed a moderate inverse correlation with the extent of PASI improvement (r = −0.25). A similar trend was observed for the IL-17/IL-13 ratio, which was also significantly lower in SR than NSR patients at baseline (median 2.47 vs. 13.27, *p* = 0.0044). This ratio yielded an AUC of 0.89 and an optimal cut-off of 0.44. However, its correlation with PASI improvement was weaker (r = −0.13), suggesting its utility as a binary predictor of response rather than a continuous outcome measure. To internally validate the discriminatory capacity of the IL-18/IL-13 ratio, leave-one-out cross-validation was performed, yielding an AUC comparable to the original model (AUC = 1.0), suggesting relative robustness despite the limited sample size.

## 3. Discussion

Super responders (SRs) to biologic therapy in psoriasis are typically defined as patients who achieve complete or nearly complete skin clearance, most commonly operationalized as achieving a Psoriasis Area and Severity Index (PASI) of 0 at a defined timepoint, such as 12 or 16 weeks after treatment initiation [7,8]. In the present study, SRs were defined as patients reaching PASI = 0 at 12 weeks of secukinumab therapy. This phenotype is of high clinical relevance, as SRs not only experience maximal symptomatic relief but also often demonstrate prolonged disease control, improved quality of life, and may require less frequent long-term dosing or treatment switching [9,10,11,12].

Previous studies have identified several clinical and demographic predictors associated with super response to biologic therapies in psoriasis. Younger age, lower body mass index (BMI), shorter disease duration, and non-smoking status have been reported as favorable clinical characteristics linked to a higher likelihood of achieving PASI 0 [12,13,14]. Additionally, patients without psoriatic arthritis and those with lower baseline disease severity or less prior biologic exposure are more likely to exhibit super response [6]. These findings suggest that a less complex or less treatment-refractory disease phenotype may predispose patients to full clinical clearance in response to targeted immunotherapy.

While clinical predictors provide valuable insights into patient stratification, they do not fully capture the underlying immunopathological mechanisms that drive differential treatment responses. To address this gap, we aimed to identify specific immunologic processes associated with the SR phenotype by characterizing cytokine profiles before and after treatment with secukinumab. This approach offers a deeper understanding of the immune signatures that may predict or accompany complete disease resolution and supports the advancement of precision-guided therapeutic strategies in psoriasis.

It is important to note that, to date, relatively few studies have specifically explored immunologic biomarkers of treatment response in psoriasis. Most available studies have focused on analyzing serum cytokine levels and their changes following various biologic therapies [5,15]. The smallest patient cohorts investigated in this context have predominantly involved anti-IL-17 therapies [16]. Even fewer studies have examined immunologic biomarkers specifically associated with the super response phenotype [10]. Our study therefore contributes valuable and novel data to this evolving field.

The relevance of our findings becomes particularly apparent when considering the role of IFN-γ in the pathogenesis of psoriasis and the response to biologic therapy. IFN-γ is a key cytokine implicated in driving neutrophil chemotaxis to the skin, promoting vascular proliferation, and inducing keratinocyte hyperproliferation within psoriatic lesions [17]. Moreover, IFN-γ enhances the function of Th17 cells and acts synergistically with IL-17 to amplify cutaneous inflammation [18]. In a recent study, Hsieh et al. demonstrated that serum IFN-γ levels positively correlated with PASI scores and that reductions in IFN-γ were associated with clinical improvement in biologically treated patients [15]. Thus, both IFN-γ levels and their dynamic changes appear to represent meaningful biomarkers for predicting biologic therapy outcomes.

In contrast to these prior findings, we did not observe significant correlations between IFN-γ levels and PASI scores, either at baseline or post-treatment. We can only speculate that this discrepancy may reflect the prior immunosuppressive exposure to cyclosporine and/or methotrexate in our patient cohort, which may have modulated the IFN-γ axis prior to biologic therapy initiation.

Interestingly, we also observed a significant negative correlation between baseline IFN-γ and baseline IL-4 levels—a cytokine central to the Th2 response. This finding may suggest that imbalances between Th1 and Th2 pathways contribute to psoriasis pathogenesis and disease severity. Further studies are warranted to explore how this immunological balance influences therapeutic outcomes.

In this context, IL-18 emerges as another cytokine of particular interest. IL-18 is produced by various cell types, including dendritic cells, macrophages, and keratinocytes [19]. This cytokine stimulates the production of IFN-γ and, in cooperation with IL-12, IL-15, and IL-23, promotes the differentiation of naïve T lymphocytes into Th1 and Th17 effector cells [20]. Elevated serum IL-18 levels have been reported in patients with psoriasis compared to healthy individuals [21]. Notably, these levels show a positive correlation with disease severity, as measured by the PASI. Animal model studies have demonstrated that IL-18 is a key mediator driving psoriatic epidermal hyperplasia and sustaining chronic psoriatic inflammation [19]. Of particular interest, IL-18 production by keratinocytes occurs constitutively, independent of active inflammation [19]. However, increased secretion of this cytokine is triggered by inflammasome activation.

Recent studies have shown that IL-18 levels remain significantly elevated in psoriasis patients even after clinically effective therapy [5]. This observation suggests that, in contrast to IL-17, IL-23, or TNF-α, IL-18 may serve as a more stable marker of the chronic inflammatory phenotype in psoriasis.

In our study, we demonstrate for the first time that lower baseline serum IL-18 levels in patients with psoriasis may predict an excellent clinical response to biologic therapy with secukinumab, defined as complete skin clearance (PASI = 0) at three months. This finding raises the hypothesis that higher baseline IL-18 levels and persistent IL-18 activity may reflect a deeper inflammatory phenotype or ongoing inflammasome activation in patients with suboptimal treatment responses. Further research is warranted to validate these observations. In particular, it would be valuable to determine whether elevated baseline IL-18 activity might predict future secondary loss of response to biologic therapy in psoriasis.

Our findings also align with the emerging understanding that psoriasis is not solely driven by Th1 and Th17 responses but also involves the Th2 axis. Similar to the findings reported by Hsieh et al. [15], we observed that an imbalance between Th1 and Th2 responses may exacerbate psoriatic symptoms. As proposed previously, an effective therapeutic strategy in psoriasis may not only involve suppression of Th1 and Th17 responses but also aim to restore immune balance by enhancing Th2-mediated pathways, particularly through cytokines such as IL-4 and IL-13 [22,23].

Supporting this concept, we observed that super responders (SRs) exhibited significantly higher baseline serum levels of IL-13 compared to non-super responders (NSRs). Moreover, the IL-18/IL-13 ratio demonstrated greater discriminatory power between SRs and NSRs than either cytokine alone. This finding further underscores the potential importance of achieving immune balance between Th1 and Th2 responses in determining super responder status. Despite the limited sample size, the robustness of the IL-18/IL-13 ratio as a predictive marker was supported by leave-one-out cross-validation, which yielded an AUC of 1.0. This result indicates the strong internal consistency of the model and highlights the potential clinical utility of this cytokine ratio for stratifying patients likely to achieve complete disease clearance.

While the immunologic role of IL-13 in psoriasis remains less clearly defined than that of Th1 or Th17 cytokines, IL-13 is known to exert anti-inflammatory and tissue-regulatory effects, including suppression of proinflammatory mediators and promotion of epithelial repair [24]. Additionally, serum IL-13 levels and IL-13-related cytokine ratios (e.g., IFN-γ/IL-13, IL-17A/IL-13) have been associated with clinical responses to biologic therapies in psoriasis [15]. These observations suggest that the elevated baseline IL-13 in super responders may reflect a Th2-skewed immunophenotype conducive to complete clearance with IL-17A blockade, although mechanistic confirmation is needed through additional studies.

Our findings suggest that IL-18 expression may be partially driven by systemic monocytic activity. We observed significant positive correlations between IL-18 concentrations (both at baseline and post-treatment) and peripheral blood monocyte counts, with the strongest association noted after therapy. Furthermore, changes in IL-18 levels (delta IL-18) also correlated with baseline monocyte levels. These findings indicate that reductions in IL-18 among super responders may, at least in part, reflect a decrease in monocyte-driven inflammation. This aligns with our earlier study, in which lower baseline monocyte counts were independently associated with achieving PASI ≤ 2 after six months of secukinumab treatment [25].

In contrast, IL-17 and IL-23 levels—although closely linked within the psoriatic cytokine axis—did not display a strong correlation in our cohort, either before or after treatment. The weak and non-significant relationship between these cytokines suggests that, at least in this patient population, their expression patterns may be regulated independently. This may reflect varying cellular sources, the compartmentalization of cytokine activity, or temporal dissociation in their regulation, despite their functional interplay in psoriatic inflammation.

Together, these exploratory findings provide mechanistic support for the observed cytokine dynamics in super responders and highlight the importance of monocyte-driven IL-18 signaling in modulating the treatment response to IL-17A blockade.

This study has several limitations that should be acknowledged. First, although the overall patient cohort was relatively small and included only nine super responders, it represents one of the larger cytokine-based investigations focused exclusively on patients treated with a single biologic agent—secukinumab—ensuring treatment homogeneity. Nonetheless, the limited sample size and subgroup imbalance may have affected the statistical power of some analyses. Second, all patients were recruited from a single clinical center. While this ensured uniform clinical management and data collection, it may limit the generalizability of the findings. Therefore, future multicenter studies with larger and more diverse populations are warranted to validate and expand upon these observations. Third, differences in national treatment guidelines and eligibility criteria for initiating biologic therapy should be considered when comparing results across studies. In Poland, biologic therapy can only be initiated following documented failure of at least three months of at least two conventional systemic treatments. Although this interval likely minimizes acute pharmacologic effects, residual immune modulation may persist. Cyclosporine has been shown to significantly reduce serum concentrations of IL-23 and TNF-α after eight weeks of therapy in patients with palmoplantar pustulosis, highlighting its suppressive effects on the IL-23 pathway [26,27]. Similarly, methotrexate decreases the frequency of pro-inflammatory, skin-homing CLA^+^ T cells—key producers of IL-17 and IL-22—with immunomodulatory effects lasting several weeks post-treatment. These findings support the hypothesis that prior immunosuppressive therapies may have influenced baseline cytokine expression in our cohort, particularly within the IL-17/IL-23/IL-18 axis. Given the limited sample size, subgroup comparisons based on prior treatment exposure were not feasible in this study. Nevertheless, future multicenter studies should include stratification by previous systemic therapies and examine the duration and immunological effects of washout periods to more precisely evaluate residual treatment impact. Finally, our analysis focused on treatment with secukinumab, an IL-17A inhibitor. While the insights gained provide valuable information on the immunologic correlates of response to this agent, further comparative studies evaluating immunologic predictors of response across different biologic classes will be important to determine the broader applicability of our findings.

## 4. Materials and Methods

### 4.1. Study Design and Participants

This prospective observational study enrolled adult patients with moderate-to-severe plaque psoriasis who initiated their first biological therapy with secukinumab. Eligibility criteria included a Psoriasis Area and Severity Index (PASI) score ≥10 and a documented lack of response, contraindication, or intolerance to at least two prior conventional systemic therapies (i.e., cyclosporine, acitretin, or methotrexate). Importantly, to minimize potential confounding effects from previous treatments on immunologic parameters, only patients in whom the interval between discontinuation of prior conventional systemic therapy and initiation of biologic treatment was at least 12 weeks were eligible for inclusion. The study was approved by the Bioethics Committee of the Medical University of Silesia in Katowice (approval number: BNW/NWN/0052/KB1/31/II/24), and written informed consent was obtained from all participants.

The following exclusion criteria were applied: age <18 years, erythrodermic psoriasis, pustular psoriasis, psoriatic arthritis, isolated nail psoriasis, genital involvement, active infections including HIV, HCV, HBV, a positive QuantiFERON-TB Gold test, and coexisting autoimmune or inflammatory disorders, including inflammatory bowel disease, rheumatoid arthritis, or spondyloarthropathies.

### 4.2. Treatment Regimen

Secukinumab was administered according to the Summary of Product Characteristics (SmPC): 300 mg subcutaneously at weeks 0, 1, 2, 3, and 4, followed by 300 mg every four weeks thereafter.

### 4.3. Data Collection and Sample Handling

At baseline (prior to treatment initiation), peripheral blood samples were collected for routine laboratory analysis and cytokine measurements. Plasma samples were stored at −80 °C until analysis. Clinical and demographic data were also collected, including age, sex, body mass index (BMI), psoriasis duration, PASI score, Dermatology Life Quality Index (DLQI), and type of previous conventional therapy. At week 12 of treatment, response to secukinumab was assessed using the PASI score. Based on this assessment, patients were stratified into two subgroups: (1) super responders (SRs), defined as PASI = 0 at week 12; and (2) non-super responders (NSRs), defined as PASI > 0 at week 12. Blood samples were collected at this timepoint for repeated cytokine quantification. Statistical analysis was subsequently conducted between the SR and NSR subgroups.

### 4.4. Cytokine Quantification

Serum concentrations of the following cytokines were measured both at baseline and after 12 weeks of treatment: IL-2, IL-4, IL-5, IL-6, IL-9, IL-10, IL-12, IL-13, IL-15, IL-17, IL-18, IL-22, IL-23, TNF-α, IFN-α, IFN-γ, IL-1α, and IL-1β. Measurements were performed using a Luminex^®^ High Performance Assay (R&D Systems, Bio-Techne, Minneapolis, MN, USA). The lower limits of detection (LOD) for each analyte were as follows: 0.1 pg/mL for IL-2, IL-4, IL-5, IL-6, IL-9, IL-10, IL-12, IL-13, IL-18, IL-22, TNF-α, IFN-α, IL-1α, and IL-1β; 0.5 pg/mL for IL-15 and IL-17; 7 pg/mL for IFN-γ; and 34 pg/mL for IL-23.

### 4.5. Statistical Analysis

Statistical analyses were conducted using R software (version 4.3.2; R Foundation for Statistical Computing, Vienna, Austria). Categorical variables were expressed as absolute numbers and percentages and compared between groups using Fisher’s exact test. For categorical comparisons involving small sample sizes, Fisher’s exact test was used to calculate *p*-values, which reflect differences in distribution rather than direct comparison of the absolute counts. Continuous variables were summarized as medians with minimum and maximum (min-max) values and compared using the Mann–Whitney U test. Changes in cytokine concentrations between baseline and post-treatment were analyzed by calculating delta values (post-treatment minus baseline). Between-group comparisons of delta values were performed using the Mann–Whitney U test.

Associations between clinical parameters (including PASI and DLQI scores) and cytokine concentrations were evaluated using Spearman’s rank correlation coefficient. Correlation matrices were visualized with heatmaps. Statistical significance was set at *p* < 0.05.

To identify cytokines predictive of super response (PASI = 0 at week 12), univariate and multivariate logistic regression models were fitted. Additionally, L1-regularized logistic regression (Lasso) was applied for variable selection using the glmnet package. The discriminative performance of models was assessed by the area under the receiver operating characteristic curve (AUC), which was computed with the pROC package. Optimal cut-off values for individual cytokines and cytokine ratios were determined based on maximization of Youden’s index. All statistical analyses and visualizations were performed using the stats, glmnet, pROC, and ggplot2 (or pheatmap) packages in R (version 4.3.2). Two-tailed *p*-values < 0.05 were considered statistically significant.

## 5. Conclusions

This study identified distinct immunologic signatures that may predict super response to secukinumab in patients with moderate-to-severe psoriasis. Lower baseline IL-18 levels, higher IL-13 levels, and a favorable IL-18/IL-13 ratio emerged as promising biomarkers associated with complete skin clearance. Our findings also highlight the relevance of immune balance between the Th1 and Th2 pathways in modulating treatment outcomes.

Importantly, these results expand the current understanding of cytokine dynamics in biologic therapy and offer a foundation for developing precision-guided treatment strategies. Although further validation in larger, multicenter cohorts and across different biologic classes is required, integrating such immunologic markers into clinical practice holds the potential to improve patient stratification, optimize therapeutic decision making, and ultimately enhance long-term disease control in psoriasis.

## Figures and Tables

**Figure 1 ijms-26-06432-f001:**
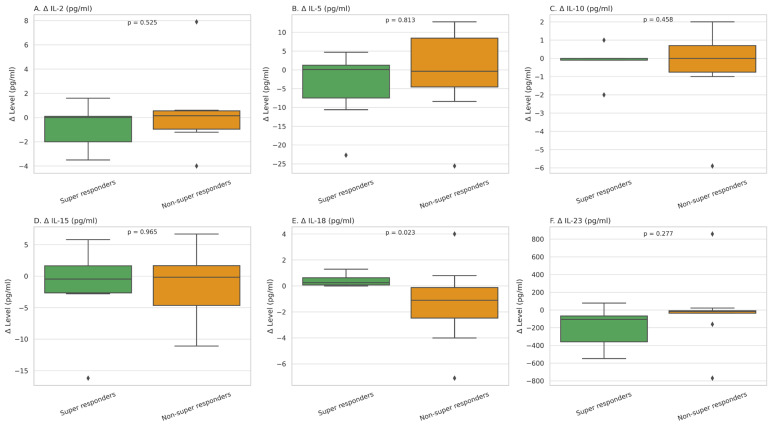
Changes in serum cytokine levels (pg/mL) (Δ, post-treatment minus baseline) for IL-2, IL-5, IL-10, IL-15, IL-18, and IL-23 in patients with psoriasis, stratified by clinical response. Panels (**A**–**F**) show the distribution of cytokine level changes among super responders (green; PASI = 0 at 3 months) and non-super responders (orange; PASI > 0). Horizontal lines indicate medians, boxes represent interquartile ranges, and whiskers show the full data range. *p*-values were calculated using the Mann–Whitney U test to compare distributions between groups. A significant difference was observed for IL-18 (panel (**E**)), with super responders showing a more pronounced decrease in IL-18 levels after treatment.

**Figure 2 ijms-26-06432-f002:**
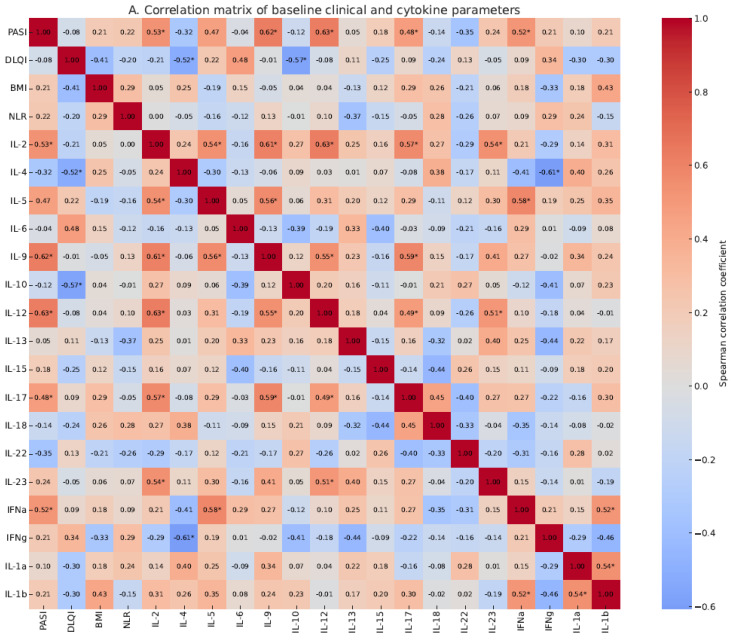
Spearman correlation matrices between clinical parameters and cytokine concentrations. (**A**) Correlations between baseline clinical and laboratory parameters (including PASI, DLQI, BMI, and NLR) and baseline cytokine levels. (**B**) Correlations between post-treatment clinical parameters (including PASI and DLQI) and post-treatment cytokine levels. Significant positive correlations are marked with an asterisk (*, *p* < 0.05). The correlation matrix heatmap was generated using the pheatmap package in R.

**Figure 3 ijms-26-06432-f003:**
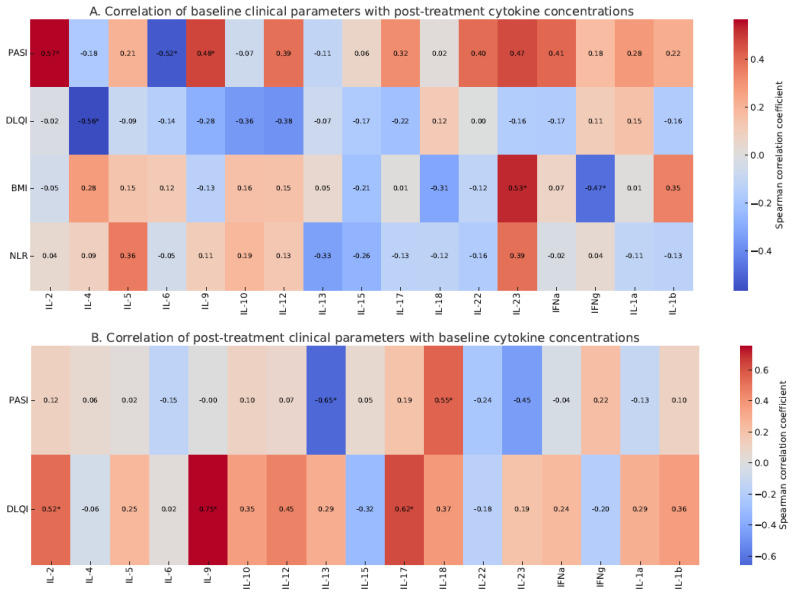
Correlation matrices illustrating cross-time associations between clinical parameters and cytokine levels. (**A**) Correlations between baseline clinical and laboratory parameters (PASI, DLQI, BMI, NLR) and cytokine levels measured after treatment. (**B**) Correlations between post-treatment clinical parameters (PASI, DLQI) and baseline cytokine levels. Significant correlations are marked with an asterisk (*, *p* < 0.05). Correlation matrix heatmap generated using the pheatmap package in R.

**Table 1 ijms-26-06432-t001:** Demographic and clinical characteristics of the study population at baseline.

	Super Responders (SRs), *n* = 9	Non-Super Responders (NSRs), *n* = 19	*p*-Value
females; *n* (%)	4 (44.4)	9 (47.4)	1.0
males; *n* (%)	5 (55.6)	10 (52.6)	1.0
age	30.50 (19.00–57.00)	45.00 (27.00–57.00)	0.15
BMI	23.66 (22.34–27.47)	26.20 (20.83–29.97)	0.41
baseline PASI	35.3 (16.2–50.1)	24.2 (20–40)	0.76
post-treatment PASI	0.0 (0.0–0.0)	6.6 (2.4–18.0)	0.0002
disease duration; months	32.1 (18–108)	37.9 (12–144)	0.54
previous treatment with cyclosporine; *n* (%)	8 (88.9)	17 (89.4)	1.0
previous treatment with methoretxate; *n* (%)	9 (100)	16 (84.2)	0.53
previous treatment with acitretin; *n* (%)	3 (33.3)	6 (31.6)	1.0
baseline CRP, mg/dL	8.25 (3.80–15.90)	10.40 (4.30–22.10)	0.63
baseline lymphocytes count; ×10^3^/μL	3.05 (2.02–4.02)	2.83 (1.56–3.55)	0.23
baseline monocytes count; ×10^3^/μL	0.41 (0.12–0.81)	0.67 (0.33–0.78)	0.03
baseline neutrophils count; ×10^3^/μL	5.22 (2.14–7.05)	5.44 (4.12–8.77)	0.59
baseline NLR	1.7 (0.9–3.1)	2.1 (1.4–5.6)	0.31

Categorical variables are presented as *n* (%) and continuous variables as median (min–max). *p*-values were calculated using Fisher’s exact test for categorical variables and the Mann–Whitney U test for continuous variables. Statistical significance was defined as *p* < 0.05. BMI; body mass index, PASI; Psoriasis Area Severity Index, CRP; C-reactive protein, NLR; neutrophil to lymphocyte ratio.

**Table 2 ijms-26-06432-t002:** Baseline and post-treatment serum cytokine levels in super responders (SRs) and non-super responders (NSRs) to secukinumab treatment of psoriasis.

	Baseline Levels			Post-Treatment Levels		
	Super Responder (SR) Group (*n* = 9)	Non-Super Responder (NSR) Group (*n* = 19)	*p*-Value	Super Responder (SR) Group (*n* = 9)	Non-Super Responder (NSR) Group (*n* = 19)	*p*-Value
IL-2; pg/mL	2.3 (0.1–17.4)	3.1 (0.1–16.9)	1.0	0.2 (0.1–20.9)	4.0 (0.1–13.8)	0.30
IL-4; pg/mL	0.1 (0.1–67.9)	0.1 (0.1–14.8)	0.6	0.1 (0.1–45.2)	0.1 (0.1–10.3)	0.34
IL-5; pg/mL	2.6 (0.1–23.1)	0.45 (0.1–34.9)	0.75	0.8 (0.1–8.1)	7.05 (0.1–15.8)	0.2
IL-6; pg/mL	0.2 (0.1–11.9)	0.1 (0.1–0.9)	0.35	0.1 (0.1–0.4)	0.1 (0.1–0.3)	1.0
IL-9; pg/mL	0.1 (0.1–4.5)	0.1 (0.1–5.2)	0.75	0.1 (0.1–5.9)	0.1 (0.1–3.1)	1.0
IL-10; pg/mL	0.2 (0.1–3.2)	0.15 (0.1–14.5)	1.0	0.25 (0.1–3.1)	0.7 (0.1–13.5)	0.34
IL-12; pg/mL	1.5 (0.1–3.7)	1.2 (0.1–4.1)	0.93	0.9 (0.1–2.2)	1.05 (0.1–3.1)	0.53
IL-13; pg/mL	1.2 (0.6–1.6)	0.35 (0.1–1.1)	0.002	0.8 (0.0–1.7)	0.35 (0.1–1.1)	0.17
IL-15; pg/mL	5.8 (0.5–55.4)	7.1 (0.1–45.2)	0.96	8.1 (0.1–39.2)	7.5 (0.4–34.1)	1.0
IL-17; pg/mL	3.3 (0.4–11.4)	4.0 (0.5–14.9)	0.35	2.7 (0.5–10.2)	3.1 (1.1–19.3)	0.42
IL-18; pg/mL	0.15 (0.1–0.4)	2.05 (0.2–8.1)	0.007	0.5 (0.1–1.7)	0.25 (0.1–11.2)	0.82
IL-22; pg/mL	0.1 (0.1–0.3)	0.1 (0.1–0.1)	0.31	0.1 (0.1–0.4)	0.1 (0.1–0.1)	0.31
IL-23; pg/mL	345.0 (98.0–1004.0)	129.0 (34.0–871.0)	0.12	145.0 (6.0–801.0)	121.0 (32.0–892.0)	0.92
TNF-α; pg/mL	3.1 (0.1–9.1)	1.1 (0.1–5.1)	0.2	2.1 (0.1–3.9)	0.4 (0.1–2.9)	0.26
IFN-α; pg/mL	0.3 (0.1–1.2)	0.25 (0.1–0.5)	0.68	0.25 (0.1–1.9)	0.2 (0.1–2.2)	0.93
IFN-γ; pg/mL	35.5 (7.0–118.0)	71.5 (7.0–142.0)	0.37	52.0 (28.0–145.0)	28.0 (7.0–88.0)	0.04
IL-1α; pg/mL	0.25 (0.1–1.8)	0.1 (0.1–4.3)	0.51	0.1 (0.1–0.9)	0.1 (0.1–0.4)	0.91
IL-1β; pg/mL	0.2 (0.1–1.1)	0.3 (0.1–1.6)	0.70	0.2 (0.1–1.4)	0.1 (0.1–1.1)	0.37

Variables are presented as median (min–max). *p*-values were calculated using the Mann–Whitney U test for continuous variables. Statistical significance was defined as *p* < 0.05.

**Table 3 ijms-26-06432-t003:** Cytokine ratios at baseline and after treatment in super responders (SRs) and non-super responders (NSRs) to secukinumab treatment of psoriasis.

	Baseline Levels			Post-Treatment Levels		
	Super Responder (SR) Group (*n* = 9)	Non-Super Responder (NSR) Group (*n* = 19)	*p*-Value	Super Responder (SR) Group (*n* = 9)	Non-Super Responder (NSR) Group (*n* = 19)	*p*-Value
IL-17/IL-4; pg/mL	27.5 (0.01–114.0)	22.3 (0.05–128.0)	1.0	21.5 (0.1–102.0)	20.5 (0.2–147.0)	0.76
IL-17/IL-13; pg/mL	2.5 (0.36–10.36)	13.3 (1.7–53.0)	0.004	3.1 (0.56–inf)	14.9 (1–96.3)	0.17
IL-23/IL-4; pg/mL	1965.0 (8.3–10040)	650.0 (2.3–5610.0)	0.11	1450.0 (0.62–8010.0)	430.0 (20.9–3990.0)	0.39
IL-23/IL-13; pg/mL	337.5 (80.6–912.7)	231.0 (90–5620.0)	0.74	270.8 (12–inf)	356.7 (80.0–3990.0)	0.54
IL-18/IL-13; pg/mL	0.18 (0.06–0.33)	3.46 (1.5–30.0)	0.00001	0.76 (0.2–inf)	2.0 (0.09–37.3)	0.62

Variables are presented as median (min-max). *p*-values were calculated using the Mann–Whitney U test for continuous variables. Statistical significance was defined as *p* < 0.05.

## Data Availability

The data that support the findings of this study are available from the corresponding author upon reasonable request.

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
