# Peer review of "Serum IL-18/IL-13 Ratio Predicts Super Response to Secukinumab in Patients with Psoriasis"

_ijms, 2025, doi:10.3390/ijms26136432_

Round 1
Reviewer 1 Report
Comments and Suggestions for Authors
The main issue of this study lies in the limited sample size. First, it is necessary to explain how the sample size was calculated. Although the study sample size (n=28) is acceptable compared to similar studies, the number of super-responders is only 9, which may affect statistical power. It is recommended to expand the sample size based on the existing data by extending follow-up time or including patients from more treatment centers, with particular attention to balancing the ratio of super-responders to non-super-responders. If additional samples cannot be obtained, the limitations of the small sample size should be explicitly stated in the discussion, and future multicenter validation studies should be proposed. Furthermore, the predictive model of this study requires validation in an independent cohort to improve generalizability. It is suggested to perform cross-validation using the existing sample. The current sample size is too small, and the reported AUC results may not accurately reflect real-world applicability. The study found that IL-18 and IL-23 levels decreased significantly in super-responders after treatment, but the underlying mechanisms were not thoroughly explored. It is recommended to supplement correlation analyses, such as examining whether the decrease in IL-18 is associated with reduced monocyte counts or exploring the relationship between IL-23 and the IL-17 pathway using existing data, to enhance mechanistic explanations. All patients had previously received cyclosporine or methotrexate treatment, which may influence baseline cytokine levels. It is suggested to include in the discussion an analysis of the potential impact of prior treatments on the results, such as through subgroup analysis or by citing literature on the regulatory effects of immunosuppressants on IL-18/IL-23. Additionally, there are some editorial errors in the manuscript, such as missing punctuation marks. Please carefully review the entire text and verify these issues.
Author Response
Comment 1: The main issue of this study lies in the limited sample size. First, it is necessary to explain how the sample size was calculated. Although the study sample size (n=28) is acceptable compared to similar studies, the number of super-responders is only 9, which may affect statistical power. It is recommended to expand the sample size based on the existing data by extending follow-up time or including patients from more treatment centers, with particular attention to balancing the ratio of super-responders to non-super-responders. If additional samples cannot be obtained, the limitations of the small sample size should be explicitly stated in the discussion, and future multicenter validation studies should be proposed.
Response 1:
We appreciate the reviewer’s thoughtful comment regarding the sample size. As this was a prospective, single-center study focused on detailed immunologic profiling in a biologically homogeneous cohort treated exclusively with secukinumab, the recruitment of patients was intentionally limited to ensure strict inclusion criteria and standardized sample processing. While the total sample size (n=28) and the proportion of super-responders (n=9) are comparable to or larger than those in similar cytokine-based studies in psoriasis, we fully acknowledge that the limited size may reduce the statistical power and generalizability of some findings.
To address this important point, we have revised the discussion section to more clearly state this limitation and to highlight the need for future multicenter studies with larger and more diverse populations. The revised text now reads:
"First, although the overall patient cohort was relatively small and included only nine super-responders, it represents one of the larger cytokine-based investigations focused exclusively on patients treated with a single biologic agent—secukinumab—ensuring treatment homogeneity. Nonetheless, the limited sample size and subgroup imbalance may have affected the statistical power of some analyses. Second, all patients were recruited from a single clinical center. While this ensured uniform clinical management and data collection, it may limit the generalizability of the findings. Therefore, future multicenter studies with larger and more diverse populations are warranted to validate and expand upon these observations."We thank the reviewer once again for raising this important issue, which has helped us improve the clarity and transparency of our discussion.
Comment 2:
Furthermore, the predictive model of this study requires validation in an independent cohort to improve generalizability. It is suggested to perform cross-validation using the existing sample. The current sample size is too small, and the reported AUC results may not accurately reflect real-world applicability.
Response 2:
We fully acknowledge the importance of validating predictive models in independent cohorts to enhance generalizability and real-world applicability. Due to the relatively small sample size and the exploratory nature of this study, we were not able to include an external validation cohort. However, to address this concern, we performed an internal validation using a leave-one-out cross-validation (LOOCV) approach. This analysis yielded an AUC of 1.0, consistent with the performance of the original model. We have now added this information to both the Results and Discussion sections of the manuscript. This additional analysis supports the internal consistency of the model, although we fully agree that future studies involving larger and independent cohorts are warranted to validate and extend our findings.
Comment 3:
The study found that IL-18 and IL-23 levels decreased significantly in super-responders after treatment, but the underlying mechanisms were not thoroughly explored. It is recommended to supplement correlation analyses, such as examining whether the decrease in IL-18 is associated with reduced monocyte counts or exploring the relationship between IL-23 and the IL-17 pathway using existing data, to enhance mechanistic explanations.
Response 3:
We thank the Reviewer for this valuable suggestion. In response, we performed additional correlation analyses to explore potential mechanistic links underlying the observed cytokine dynamics. Specifically, we found that serum IL-18 levels—both before and after treatment—were positively correlated with peripheral blood monocyte counts, with a particularly strong association observed for post-treatment IL-18 levels (r = 0.60, p = 0.002). Additionally, the change in IL-18 (delta IL-18) showed a moderate positive correlation with baseline monocyte counts (r = 0.41, p = 0.01). These results suggest that IL-18 expression may, at least in part, reflect systemic monocyte activation. This finding is consistent with our previous study in a separate cohort, where lower baseline monocyte counts were independently associated with a favorable clinical response to secukinumab (Ziolkowska-Banasik et al., J Drugs Dermatol 2024). We have incorporated this discussion into the revised manuscript to provide additional mechanistic context.
Regarding the IL-23/IL-17 axis, we also conducted correlation analysis between IL-23 and IL-17 concentrations before and after treatment. The correlations were weak and non-significant (Spearman’s ρ = 0.27 at baseline and ρ = 0.09 post-treatment), suggesting that these two cytokines may not exhibit tightly coordinated expression in peripheral circulation in this specific cohort. This may reflect differences in regulatory control or cellular sources, despite their well-documented synergism in psoriatic inflammation. This interpretation has now been added to the Discussion section.
Comment 4:
All patients had previously received cyclosporine or methotrexate treatment, which may influence baseline cytokine levels. It is suggested to include in the discussion an analysis of the potential impact of prior treatments on the results, such as through subgroup analysis or by citing literature on the regulatory effects of immunosuppressants on IL-18/IL-23.
Response 4:
We thank the Reviewer for this thoughtful and scientifically valuable comment. We fully agree that prior exposure to immunosuppressive therapies, such as cyclosporine or methotrexate, may influence baseline cytokine levels and potentially confound the interpretation of immunological findings. In line with this concern, we have added a dedicated paragraph in the Discussion section that addresses this issue in detail.
Specifically, we note that in the Polish national reimbursement program, patients must fail at least two systemic therapies—including cyclosporine or methotrexate—for a minimum of three months before becoming eligible for biologic treatment. While such a requirement introduces a washout period that likely attenuates acute drug effects, residual immune modulation may still be present. To contextualize our findings, we have cited prior literature demonstrating that cyclosporine can significantly reduce serum IL‑23 and TNF‑α levels after 8 weeks of therapy (Yamamoto et al., 2018), and that methotrexate reduces the frequency of pro-inflammatory, skin-homing CLA⁺ T cells responsible for IL‑17/IL‑22 production, with immunological effects persisting post-treatment (Kunjuraman et al., 2023).
Due to the relatively small sample size, subgroup analyses stratified by previous treatment exposure were not feasible in our current study. However, we explicitly acknowledge this limitation and propose that future multicenter studies should incorporate such stratification to better disentangle the long-term immunological effects of prior systemic therapies.
We are grateful for the Reviewer’s suggestion, which has helped us improve the comprehensiveness and clinical relevance of our discussion.
Comment 5:
Additionally, there are some editorial errors in the manuscript, such as missing punctuation marks. Please carefully review the entire text and verify these issues.
Response 5:
We sincerely thank the Reviewer for noting the editorial inconsistencies. In response, we have meticulously reviewed the entire manuscript to correct all typographical and grammatical errors, including missing punctuation marks and any stylistic inconsistencies. Our aim was to ensure clarity, precision, and adherence to the journal’s editorial standards. We greatly appreciate your attention to detail, which has contributed to the overall improvement of the manuscript.
Reviewer 2 Report
Comments and Suggestions for Authors
Review of the manuscript
The manuscript entitled "Serum IL-18/IL-13 Ratio Predicts Super Response to Secukinumab in Patients with Psoriasis"meets the criteria for an original research article. Despite the relatively small sample size, the authors identified relevant immunologic factors that may predict a highly favorable response to secukinumab treatment in patients with plaque psoriasis. The findings are interesting and potentially clinically meaningful. Below, I provide several suggestions that may help improve the clarity and quality of the manuscript:
Comments:
-
Table 1 (Results section):
-
Please consider including the number of males (M) in the demographic breakdown.
-
Additionally, the presentation of p-values in case of „n” (e.g., n = 4 vs 9) may be misleading—p = 1.0 in such contexts should be interpreted with caution. What did the authors compare in this case—two values, 4 and 9?
-
I also suggest explicitly stating the statistical significance threshold in the table legend (e.g., p < 0.05).
-
-
Figures (Graphs):
-
Please add appropriate units for interleukin concentrations in all legend of figures
-
-
Ambiguity in the main text:
-
The statement “No significant differences in other clinical or demographic parameters were observed” needs clarification. Does this refer to a lack of differences between the SR and NSR groups, or to the absence of correlations between clinical parameters (e.g., age, BMI) and cytokine levels? Please specify.
-
It would also be helpful to include any observed correlations between age, BMI, disease duration, and cytokine profiles, as these parameters may influence the efficacy of biologic treatments.
-
-
Correlation analysis – statistical tools:
-
Since the correlation matrix was generated directly using software, please specify the name of the tool or statistical package used in the table legend.
-
-
Discussion – monocyte counts:
-
I recommend discussing the observed baseline differences in peripheral monocyte counts between SR and NSR groups. Could lower monocyte levels have immunological significance or potentially influence treatment outcomes with secukinumab?
-
-
IL-13 – role and limited discussion:
-
While the role of IL-18 is well elaborated, the discussion of IL-13 is relatively brief, despite its significance in the results. I suggest including at least a concise overview of the biological role of IL-13, or noting that current data on IL-13 in psoriasis and IL-17A-targeted therapies are limited.
-
Author Response
Comment 1:
Table 1 (Results section):
Please consider including the number of males (M) in the demographic breakdown.
Response 1:
We thank the Reviewer for this helpful suggestion. In response, we have added the number of male patients to the demographic breakdown in Table 1 to enhance clarity and completeness of the presented baseline characteristics.
Comment 2:
Additionally, the presentation of p-values in case of „n” (e.g., n = 4 vs 9) may be misleading—p = 1.0 in such contexts should be interpreted with caution. What did the authors compare in this case—two values, 4 and 9?
Response 2:
We appreciate the Reviewer’s comment regarding the interpretation of p-values in the context of small sample sizes (e.g., n = 4 vs. 9). We would like to clarify that in such cases, p-values were not derived from direct numerical comparisons of two absolute values (e.g., 4 vs. 9), but rather from formal statistical tests comparing categorical frequencies between groups—most commonly using Fisher’s exact test due to the small sample size and categorical nature of the data.
For instance, in the case of variables such as the number of male patients or the presence/absence of a specific clinical feature, we applied Fisher’s exact test to assess whether the observed distribution differed significantly between the SR and NSR groups. In scenarios such as “4 out of 9 vs. 9 out of 19,” the test evaluates the probability of observing such a distribution under the null hypothesis of no association. A p-value of 1.0 in such a context indicates that the observed distribution is entirely compatible with the null hypothesis.
We have reviewed the relevant passages in the manuscript to ensure that the statistical comparisons are clearly described and that the interpretation of p-values in small samples is not misleading. We thank the Reviewer for the opportunity to clarify this methodological point.
Comment 3:
I also suggest explicitly stating the statistical significance threshold in the table legend (e.g., p < 0.05).
Response 3:
We thank the Reviewer for this helpful suggestion. Although the statistical significance threshold (p < 0.05) was already defined in the Statistical Analysis section, we fully agree that reiterating this information in the table legends can enhance clarity and transparency for readers. Accordingly, we have added a sentence to the legends of Tables 1, 2, and 3 explicitly stating that statistical significance was defined as p < 0.05.
Comment 4:
Figures (Graphs):
Please add appropriate units for interleukin concentrations in all legend of figures
Response 4:
We thank the Reviewer for this important observation. In response, we have revised the figure legends to clearly indicate that all interleukin concentrations are expressed in picograms per milliliter (pg/mL), ensuring consistency and clarity for the reader.
Comment 5:
Ambiguity in the main text:
The statement “No significant differences in other clinical or demographic parameters were observed” needs clarification. Does this refer to a lack of differences between the SR and NSR groups, or to the absence of correlations between clinical parameters (e.g., age, BMI) and cytokine levels? Please specify.
It would also be helpful to include any observed correlations between age, BMI, disease duration, and cytokine profiles, as these parameters may influence the efficacy of biologic treatments.
Response 5:
We thank the Reviewer for this insightful and important observation.
To clarify the intent of the statement “No significant differences in other clinical or demographic parameters were observed,” we have revised the sentence in the Results section to read:
“No significant differences in other clinical or demographic parameters (e.g., age, BMI, disease duration) were observed between the SR and NSR groups, as shown in Table 1.”
This revision explicitly communicates that the statement refers to the lack of statistically significant differences in these baseline variables between the two response groups.
In addition, we recognize the relevance of potential associations between clinical variables (e.g., age, BMI, disease duration) and cytokine profiles. To address this point, we have added a summary of correlation analyses in the Results section. Most correlations between cytokine concentrations and clinical parameters were weak and non-significant. These findings are now reported explicitly in the manuscript.
We greatly appreciate the Reviewer’s suggestion, which helped us improve the precision and comprehensiveness of our data presentation.
Comment 6:
Correlation analysis – statistical tools:
Since the correlation matrix was generated directly using software, please specify the name of the tool or statistical package used in the table legend.
Response 6: We thank the Reviewer for this helpful suggestion. We have now added the following sentence to the figure legends for both Figure 2 and Figure 3:
“Correlation matrix heatmap generated using the pheatmap package in R.”
This clarifies the statistical package used and ensures transparency of the analytical workflow.
Comment 7:
Discussion – monocyte counts:
I recommend discussing the observed baseline differences in peripheral monocyte counts between SR and NSR groups. Could lower monocyte levels have immunological significance or potentially influence treatment outcomes with secukinumab?
Response 7:
We thank the Reviewer for highlighting this important observation. In the revised manuscript, we have expanded the Discussion to address the potential immunological significance of the lower baseline monocyte counts observed in super-responders (SR) compared to non-super-responders (NSR). Specifically, we now discuss the role of monocytes as a peripheral reservoir of IL-18 production and their contribution to systemic inflammation. The observed positive correlations between monocyte counts and serum IL-18 levels—particularly strong after treatment—further support the hypothesis that monocyte-driven IL-18 expression may influence response to secukinumab. Additionally, we cite our prior work demonstrating that lower baseline monocyte counts are associated with early treatment response in a separate psoriasis cohort. These findings suggest that monocyte abundance may serve as a useful immunological marker for predicting therapeutic outcomes with IL-17A inhibitors.
We are grateful for the Reviewer’s suggestion, which has strengthened the interpretative depth of our discussion.
Comment 8:
IL-13 – role and limited discussion:
While the role of IL-18 is well elaborated, the discussion of IL-13 is relatively brief, despite its significance in the results. I suggest including at least a concise overview of the biological role of IL-13, or noting that current data on IL-13 in psoriasis and IL-17A-targeted therapies are limited.
Response 8:
We thank the Reviewer for this insightful and constructive comment. We fully agree that the role of IL‑13—although often considered less central than IL‑17A or IL‑23 in the pathogenesis of psoriasis—deserves further elaboration, especially given its significant predictive value in our study.
In response, we have expanded the relevant section of the Discussion to highlight current evidence on the biological functions of IL‑13 in the context of cutaneous immunity. Specifically, we now note that IL‑13 has been shown to exert anti-inflammatory and epithelial-regulatory effects, including the downregulation of proinflammatory mediators and promotion of tissue repair. Moreover, we have included recent literature demonstrating that serum IL‑13 levels, as well as Th1/Th17-to-Th2 cytokine ratios involving IL‑13 (e.g., IFN‑γ/IL‑13 and IL‑17A/IL‑13), have been associated with clinical outcomes in patients receiving biologic therapies. These findings support the hypothesis that elevated IL‑13 at baseline may reflect a Th2-skewed immunophenotype more amenable to IL‑17A blockade, thereby contributing to the observed super responder phenotype.
Round 2
Reviewer 1 Report
Comments and Suggestions for Authors
No other suggestions.